# Constitutive CD8 expression drives innate CD8+ T-cell differentiation via induction of iNKT2 cells

Satoshi Kojo* ⓘ, Michiko Ohno-Oishi*, Hisashi Wada, Sebastian Nieke, Wooseok Seo ⓘ, Sawako Muroi, Ichiro Taniuchi ⓘ

**Temporal down-regulation of the CD8 co-receptor after receiving positive-selection signals has been proposed to serve as an important determinant to segregate helper versus cytotoxic lineages by generating differences in the duration of TCR signaling between MHC-I and MHC-II selected thymocytes. By contrast, little is known about whether CD8 also modulates TCR signaling engaged by the non-classical MHC-I–like molecule, CD1d, during development of invariant natural killer T (iNKT) cells. Here, we show that constitutive transgenic CD8 expression resulted in enhanced differentiation of innate memory-like CD8+ thymocytes in both a cell-intrinsic and cell-extrinsic manner, the latter being accomplished by an increase in the IL-4–producing iNKT2 subset. Skewed iNKT2 differentiation requires cysteine residues in the intracellular domain of CD8α that are essential for transmitting cellular signaling. Collectively, these findings shed a new light on the relevance of CD8 down-regulation in shaping the balance of iNKT-cell subsets by modulating TCR signaling.**

## Introduction

The thymus provides a specific microenvironment that supports development of several types of T cells, including innate-like T cells, such as invariant natural killer T (iNKT) cells and mucosal-associated invariant T (MAIT) cells, and conventional T cells. Signaling via the TCR plays a central role in driving differentiation of both innate-like and conventional T cells (Hogquist & Jameson, 2014), although the TCR diversity and the selecting MHC–presenting antigens are quite different between these two types of T cells; innate-like T cells such as iNKT cells and MAIT cells express invariant TCRs that recognize non-peptide antigens presented on non-classical MHC, CD1d and MR1, respectively (Godfrey et al, 2015), whereas conventional T cells express diverse TCRs recognizing peptide antigens presented on classical MHC (Hogquist & Jameson, 2014).

Essential roles of CD4/CD8 co-receptors in TCR/MHC interaction during differentiation of the two major conventional T-cell subsets,

CD4+ helper and CD8+ cytotoxic cells, from common precursors, CD4+CD8+ double-positive (DP) thymocytes, are well characterized. Thymocytes positively selected by class II MHC molecules (MHC-II selected thymocytes) develop into CD4+CD8− single-positive (SP) thymocytes that are committed to the helper lineage, whereas MHC-I–selected thymocytes are directed to become CD4−CD8+ SP thymocytes committed to the cytotoxic lineage (Ellmeier et al, 1999). It has been proposed that differences in the duration of the positive-selection signal instruct distinct fates in post-selection thymocytes (Singer et al, 2008). Thus, briefer TCR signals in MHC-I–selected thymocytes caused by temporal down-regulation of the CD8 co-receptor guide post-selection thymocytes to differentiate into CD4−CD8+ SP thymocytes. On the other hand, persistent TCR signals in MHC-II–selected thymocytes supported by constitutive CD4 expression activate a developmental program toward the helper-lineage T cells via induction of the zing-finger transcription factor ThPOK (He et al, 2005; Sun et al, 2005) through antagonizing a transcriptional silencer in the *Zbtb7b* gene encoding ThPOK (He et al, 2008; Setoguchi et al, 2008). Therefore, in what is called the kinetic signaling model, distinct expression kinetics between CD4 and CD8 co-receptors have been proposed to play a key role in segregating helper and cytotoxic lineages (Singer et al, 2008). In line with this model, perturbation of positive-selection signaling duration in MHC-II–selected thymocytes re-directs them to become CD8+ cytotoxic-lineage cells (Sarafova et al, 2005; Singer et al, 2008; Adoro et al, 2012). On the other hand, constitutive transgenic CD8 expression guides about 30% of MHC-I–selected thymocytes to differentiate into CD4+ cells (Bosselut et al, 2001). One proposed explanation for the low efficiency of such redirected differentiation was down-regulation of the transgenic CD8α chain that hetero-dimerized with endogenous CD8β chain.

In addition to TCR signals, cytokines play important roles in controlling T-cell differentiation in the thymus. Signals by IL-7 are crucial for the differentiation of CD8 SP thymocytes (McCaughtry et al, 2012). Recently, IL-4 has been shown to support differentiation of another type of CD8 SP thymocyte with the characteristics of both the memory and innate cells, which is referred to as innate memory-like CD8 T cells (Weinreich et al, 2010). The iNKT2 subset of iNKT cells produces IL-4 and has been shown to be a major source

Laboratory for Transcriptional Regulation, RIKEN Center for Integrative Medical Sciences, Yokohama, Japan

Correspondence: ichiro.taniuchi@riken.jp
*Satoshi Kojo and Michiko Ohno-Oishi contributed equally to this work

of IL-4 in the thymic environment. Accordingly, an increase in the numbers of iNKT2 cells, although they represent only a tiny sub-population of total thymocytes, has a significant impact on the generation of innate memory-like CD8 T cells (Lee et al, 2013). In addition to iNKT2 cells, iNKT1 cells expressing IFN-γ and iNKT17 cells expressing IL-17 are also differentiated from iNKT precursors (Constantinides & Bendelac, 2013). However, little is known about how balanced differentiation of such iNKT-cell subsets is regulated.

In this study, we generated a novel transgenic mouse model expressing the CD8αβ heterodimer or the CD8αα homodimer in the absence of endogenous CD8α/CD8β chains and MHC-II molecules and observed that two-thirds of MHC-I–selected thymocytes differentiated into CD4⁻CD8⁺ SP thymocytes, most of which acquired signatures of innate memory-like CD8 T cells in both cell-intrinsic and cell-extrinsic manner. The cell-extrinsic mechanism was linked to results from enhanced differentiation of the iNKT2-cell subset. Thus, our study sheds new light on the physiological relevance of down-regulation of the *Cd8* gene to fine-tune the balance of iNKT-cell subsets.

# Results

### Developmental pathway to CD4⁺ T cells from the CD8 SP stage in *Zbtb7b*$^{ΔTE/ΔTE}$ mice

The initial activation of the *Zbtb7b* gene upon receiving positive-selection signals is achieved mainly by a thymic enhancer (TE) (He et al, 2008; Muroi et al, 2013). Therefore, removal of the TE from the *Zbtb7b* locus results in delayed and low-level expression of ThPOK in newly selected thymocytes (Muroi et al, 2013). However, sequential activation of a proximal enhancer restores ThPOK expression in a later developmental stage (Muroi et al, 2008). The impaired ThPOK expression kinetics due to loss of the TE in *Zbtb7b*$^{ΔTE/ΔTE}$ mice results in redirected differentiation of a small proportion of MHC-II–selected thymocytes into CD4⁻CD8⁺ SP thymocytes (Muroi et al, 2013). On the other hand, enforced ThPOK expression in CD8⁺ T cells was shown to activate some helper-lineage signature genes (Jenkinson et al, 2007). Given the restored ThPOK expression in the later developmental stage of MHC-II–selected thymocytes in *Zbtb7b*$^{ΔTE/ΔTE}$ mice, we examined whether there are MHC-II–selected thymocytes from some developmental stages toward the CD8-lineage that acquire a CD4⁺CD8⁻ phenotype. We accomplished this by a fate-mapping approach using an E8I-Cre transgene, whose expression is driven by the *E8I* enhancer known to be activated specifically in mature CD4⁻CD8⁺ SP thymocytes (Ellmeier et al, 1997) and can be monitored by GFP expression from *ires-gfp* sequences present in the transgene (Seo et al, 2017). By crossing the *E8I-Cre* transgenic mice to a *Rosa26*$^{YFP}$ reporter strain, about 70% of CD8⁺ splenic T cells were marked by YFP expression, which could be distinguished from the weaker GFP signal derived from the *E8I-Cre* transgene (Fig 1A), whereas only 0.02% of CD4⁺ splenic T cells were marked by YFP. On the other hand, the percentage of YFP-positive CD4⁺ T cells was significantly increased, up to 0.07% on average, in *Zbtb7b*$^{ΔTE/ΔTE}$ mice (Fig 1A and B). This result indicates that there exists a developmental pathway toward CD4⁺ T

cells in the *Zbtb7b*$^{ΔTE/ΔTE}$ mice from a developmental stage where the *E8I-Cre* transgene is activated, which occurs after CD4 down-regulation in control mice (Seo et al, 2017). Such reversible differentiation to CD4⁺ T cells after possible loss of CD4 expression was not compatible with the kinetic signaling model, prompting us to revisit the relevance of distinct co-receptor expression kinetics in the CD4⁺ helper versus CD8⁺ cytotoxic lineage choice.

### Differentiation of innate memory-like CD8 T cells under constitutive CD8αβ expression conditions

The effect of constitutive CD8αβ co-receptor expression in vivo was examined previously in the presence of the endogenous CD8β chain. Possible down-regulation of the transgenic CD8α chain, which could dimerize with the endogenous CD8β chain, was discussed as a possible reason why most of the MHC-I–restricted cells were differentiated into CD4-negative cytotoxic T cells in this transgenic model (Bosselut et al, 2001). To analyze the effect of constitutive CD8αβ expression in the absence of both endogenous CD8α and CD8β chains, we generated a mutant *Cd8* locus, by sequential targeting in ES cells, referred to as *Cd8*$^{Δab}$, that lacks coding regions for both CD8α and CD8β chains in the *Cd8a* and *Cd8b1* genes, respectively (Figs 2A–C and S1). For transgenic expression of CD8α and CD8β chains, a cDNA encoding either chain was inserted into the *Rosa26* locus, generating a *Rosa26*$^{8a}$ and *Rosa26*$^{8b}$ locus, respectively (Fig 2A). Induction of CD8α and CD8β chains from the *Rosa26*$^{8a}$ and *Rosa26*$^{8b}$ loci upon Cre-mediated excision of the stop cassette was confirmed by flow cytometry analyses (Fig 2D). However, the expression level of transgenic CD8α, which is represented in CD4⁺ population at the lower left panel in Fig 2D was lower than that of endogenous CD8α (CD8 SP population at the upper left panel in Fig 2D), presumably because of the weak promoter activity in the *Rosa26* locus.

We next examined expression kinetics of transgenic CD8α expression in *Rosa26*$^{8a/8a}$: *Cd8*$^{Δab/Δab}$: *Cd4-Cre* mice. As expected, expression level of transgenic CD8α from the Rosa26 locus was stable and did not shown down-regulation after positive selection (Fig S2), confirming constitutive expression of CD8 co-receptor. To examine differentiation of MHC-I selected–thymocytes, we then generated *I-Ab*$^{-/-}$: *Rosa26*$^{8a/8b}$: *Cd8*$^{Δab/Δab}$: *Cd4-Cre* mice, hereafter referred to as *MHC-II*$^{O}$:*CD8ab-Tg* mice. Despite lower transgenic CD8αβ expression levels, the percentage of mature thymocytes, defined as the CD24$^{lo}$TCRβ$^{hi}$ population, was equivalent between *Wt* and *MHC-II*$^{O}$:*CD8ab-Tg* mice (Fig 3A). In the mature thymocytes population of *MHC-II*$^{O}$:*CD8ab-Tg* mice, although around 30% cells acquired CD4 expression, about 70% cells emerged as CD4-negative cells (Fig 3A). The balance of CD4⁺ and CD4⁻ cells was maintained after their egress from the thymus. In the peripheral lymphoid tissues, such as spleen, of *MHC-II*$^{O}$:*CD8ab-Tg* mice, the αβT-cell population consisted of about 65% CD4⁻ cells and 35% CD4⁺ cells (Fig S3A). This proportion of CD4⁺ to CD4⁻ cells in *MHC-II*$^{O}$:*CD8ab-Tg* mice was similar to that reported in the previous transgenic model expressing CD8α and CD8β from mini-transgenes at equivalent levels to endogenous CD8αβ (Bosselut et al, 2001), indicating that, in spite of the lower level of transgenic CD8αβ expression, our CD8αβ transgenic model is likely to mimic developmental processes in the previous model.

**Life Science Alliance**

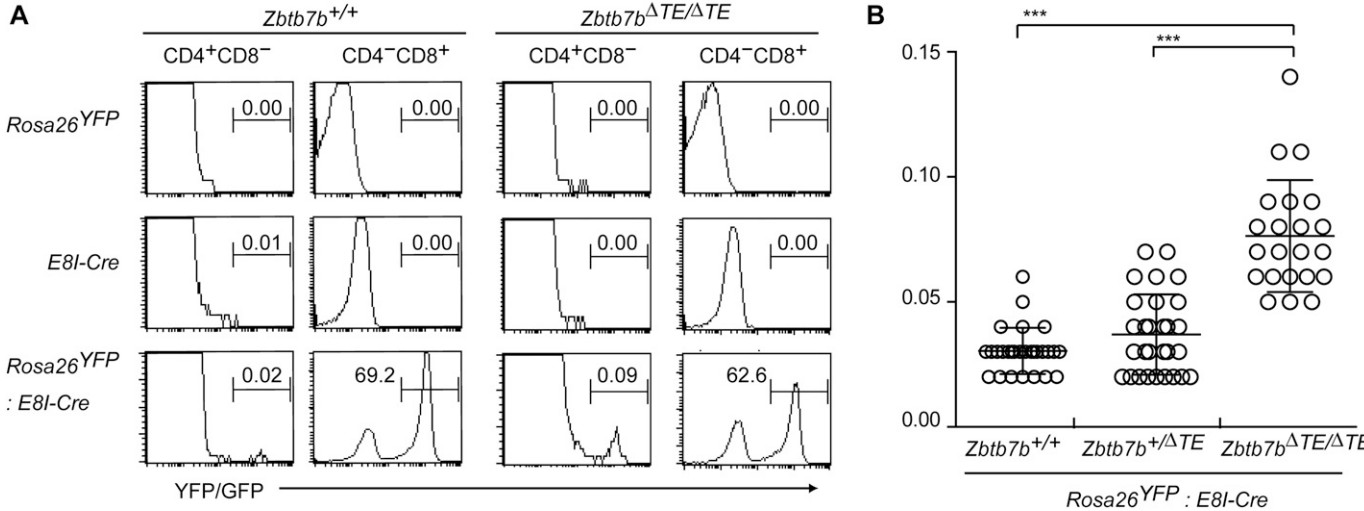

**Figure 1.  Differentiation pathway to CD4⁺ T cells after *E8I* activation in *Thpok^{DTE/DTE}* mice.**
**(A)** Histograms showing GFP expression from the *E8I-Cre* transgene and YFP expression from the *Rosa26-STOP-YFP* allele of splenic CD4⁺CD8⁻ and CD4⁻CD8⁺ T cells of mice with indicated genotypes. One representative of at least three experiments. **(B)** Graph showing summary of the percentage of YFP⁺ CD4⁺CD8⁻ splenic T cells of *Zbtb7^{+/+}*, *Zbtb7b^{+/ΔTE}*, and *Zbtb7b^{ΔTE/ΔTE}* mice that are hemizygous for E8I-Cre and *Rosa26-STOP-YFP* transgene. Mean ± SD. \*\*\**P* < 0.001 (Kruskal–Wallis test with Dunn's multiple comparisons test).

We next examined expression of Runx3, a central transcription factor for cytotoxic T-cell development, by using a *Runx3^{tdTomato}* reporter allele (Kojo et al, 2017). CD4⁻ mature thymocytes emerged in *MHC-II^o:CD8ab-Tg* mice expressed Runx3-tdTomato, albeit at a lower level than that in control CD8⁺ T cells in control mice (Fig 3A), indicating that these cells are related to the cytotoxic lineage. To further characterize the CD4⁻ mature thymocytes in *MHC-II^o:CD8ab-Tg* mice, we performed transcriptome analyses by RNA-seq. Principal component analyses indicated that CD4⁻ mature thymocytes that had differentiated in *MHC-II^o:CD8ab-Tg* mice were different from control *wild-type* CD8 SP thymocytes (Fig S3B). Analyses of differentially expressed genes revealed that *Gzmk*, *Zbtb16*, and *Tbx21* genes were up-regulated in the CD4⁻ mature thymocytes (Fig 3B). We also confirmed higher expression of Eomes, one of the signature genes for innate memory-like CD8 T cells (Weinreich et al, 2010), in CD8 SP thymocyte population of *MHC-II^o:CD8ab-Tg* mice in subsequent qRT-PCR and flow cytometry analyses (Figs 3B and S3C). CD44 and CD122 are known surface markers for innate memory-like CD8 T cells. Higher CD44 expression was detected in about half of the CD4⁻ mature thymocytes in *MHC-II^o:CD8ab-Tg* mice and CD122 expression was detected in a half of the CD44^{hi} cells (Figs 3C and S3D). An increase in CD44^{hi}CD122^{hi} cells was also observed in spleen CD4⁻ αβT-cell population (Fig S3E). These observations demonstrate that half of the CD4⁻ mature thymocytes that emerge in *MHC-II^o:CD8ab-Tg* mice are innate memory-like CD8 T cells. An increase in the CD44^{hi}CD122^{hi} subset was also induced by transgenic CD8αβ expression in an MHC-sufficient background (Fig 3C). We also found that expression of just the CD8α chain that could function as a CD8αα homodimer resulted in enhanced differentiation of CD44^{hi}CD122^{hi} innate memory-like CD8 T cells (Fig 3C). Thus, under lower but constitutive CD8αβ or CD8αα expression, more than half of the MHC-I–selected cells remain to be differentiated into CD4⁻ cytotoxic lineage–related cells with a skewed differentiation toward innate memory-like CD8⁺ cells.

## Increase in the iNKT2 subset under constitutive CD8αβ expression

During the analysis of promyelocytic leukaemia zinc finger (PLZF) expression, we noticed that around 40% of CD4⁺ mature thymocytes expressed a higher level of PLZF than that in innate memory-like CD8 T cells (Fig 4A). Innate-like CD4⁺ T cells, including CD1d-restricted iNKT cells (Kovalovsky et al, 2008) and MR1-restricted MAIT cells (Rahimpour et al, 2015), have been shown to express PLZF at high levels. Interestingly, differentiation of innate memory-like CD8 T cells in the thymus has been shown to be enhanced by IL-4 secreted by iNKT2 cells (Verykokakis et al, 2010; Weinreich et al, 2010; Lee et al, 2013). We, therefore, examined CD4⁺PLZF^{hi} cells in *MHC-II^o: CD8ab-Tg* mice in terms of iNKT characteristics, such as expression of the invariant TCR, which can be specifically detected by CD1d dimers loaded with αGalCer (αGC) (Matsuda et al, 2000). About 30% of CD4⁺ mature thymocyte of *MHC-II^o:CD8ab-Tg* mice were stained with CD1d-αGC and were positive for transgenic CD8 expression, whereas only 5% of those cells were stained in control mice (Figs 4B and S4A). Such iNKT cells were present at a similar ratio in CD4⁺ T-cell population in *MHC-II^o* mice expressing only endogenous CD8αβ (Fig 4B). On the contrary, in MHC-II–sufficient background, both frequency and numbers of iNKT cells were reduced in our *CD8ab-Tg* mice as was previously reported (Lantz & Bendelac, 1994), whereas these are restored in MHC-II–deficient background (Fig S4A). These observations indicate that iNKT cells are one of the major CD4⁺ subsets arising after elimination of MHC-II–selected thymocytes. Whereas PLZF expression was not detected in the CD4⁺CD1d-αGC⁻ population in *wild-type* control mice, there remained a small proportion of CD1d-αGC⁻PLZF⁺CD4⁺ cells in the thymus of *MHC-II^o:CD8ab-Tg* mice as well as *MHC-II^o* mice (Fig 4B).

Although the frequency of iNKT cells was similar between *MHC-II^o* and *MHC-II^o:CD8ab-Tg* mice, an increase in innate memory-like CD8⁺ T cells occurred only in *MHC-II^o:CD8ab-Tg* mice (Fig 3C). Given the important role of IL-4 to support innate memory-like CD8⁺ T-cell

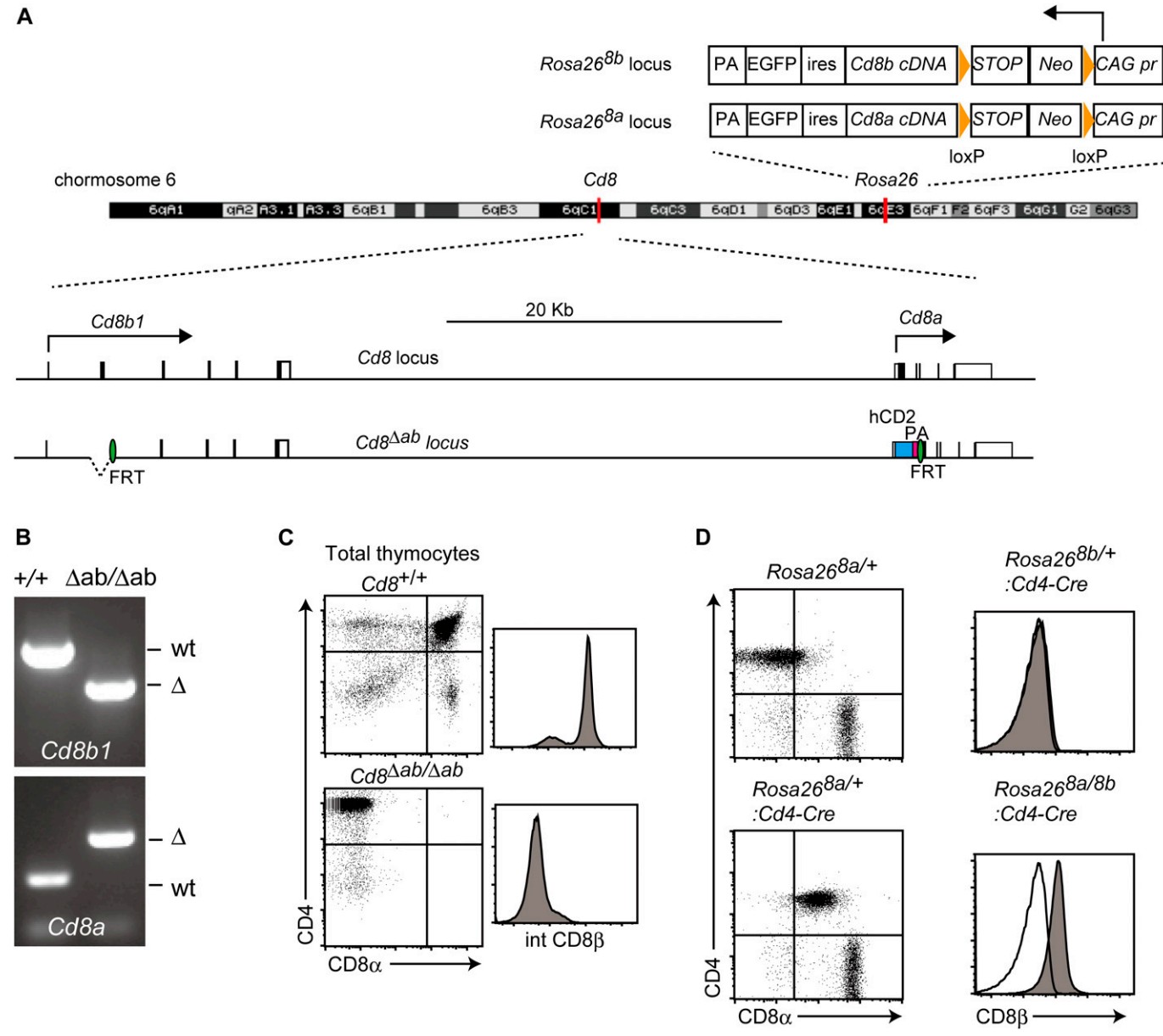

**Figure 2. Transgenic mouse line expressing CD8α and CD8β chain in the absence of endogenous CD8α and CD8β chains.**
**(A)** A scheme showing structures of *Rosa26^8a*, Rosa26^8b*, *Cd8*, and *Cd8^Δab* loci on the mouse chromosome 6. Orange triangles and green ovals represent loxP and FRT sequences, respectively. hCD2 was inserted to replace the *Cd8a* locus. Arrows indicate direction of transcription at the *Cd8* and Rosa26 loci. PA, polyA adenylation signals; *Neo*, *neomycin*-resistance gene; ires, internal ribosomal entry sites. **(B)** Gel images of genotyping PCR showing deletion of the second exon at the *Cd8b1* locus and insertion of the hCD2 cDNA at the Cd8a locus. **(C, D)** Flow cytometry analyses for CD4 and CD8a expression of total thymocytes of *Cd8^{+/+}* and *Cd8^{Δab/Δab}* mice (C) and TCRβ^+ spleen T cells of *Rosa26^{8a/+}* and *Rosa26^{8a/+}: CD4-Cre* mice (D). **(C, D)** Histogram showing intracellular staining of CD8β chain in total thymocytes of *Cd8^{+/+}* and *Cd8^{Δab/Δab}* mice (C) and surface CD8β expression of spleen CD4^+ T cells of *Rosa26^{8b/+}:CD4-Cre* and *Rosa26^{8a/8b}: CD4-Cre* mice (D). CD8β expression by wild-type CD4^+ T cells is shown as an open histogram as a negative control. One representative of two independent experiments.

differentiation (Verykokakis et al, 2010; Lee et al, 2013), we asked whether differentiation of iNKT-cell subsets is different between *MHC-II^o* and *MHC-II^o:CD8ab-Tg* mice. After CD24 down-regulation, the developmental stages of iNKT cells are divided into the CD44^−NK1.1^− stage 1, the CD44^+NK1.1^− stage 2, and the CD44^+NK1.1^+ stage 3 (Benlagha et al, 2002). In the thymus of *MHC-II^o:CD8ab-Tg* mice, proportions of CD44^−NK1.1^− stage 1 and CD44^+NK1.1^− stage 2 cells were significantly increased compared with control and *MHC-II^o* mice and, this was

accompanied by a relative decrease in CD44^+NK1.1^+ stage 3 cells (Fig 4C). The T-bet^loPLZF^hi iNKT2-cell subset is included in the stage 1 and stage 2 iNKT-cell population (Constantinides & Bendelac, 2013; Lee et al, 2013). As expected, the proportion of the T-bet^loPLZF^hi iNKT2-cell subset was significantly higher in *MHC-II^o:CD8ab-Tg* mice than in *wild-type* and *MHC-II^o* mice (Fig 4C), which was consistent with an increase in the number of iNKT2 cells in *MHC-II^o:CD8ab-Tg* mice (Fig S4B). We further confirmed an increase in the iNKT2 subset by using other

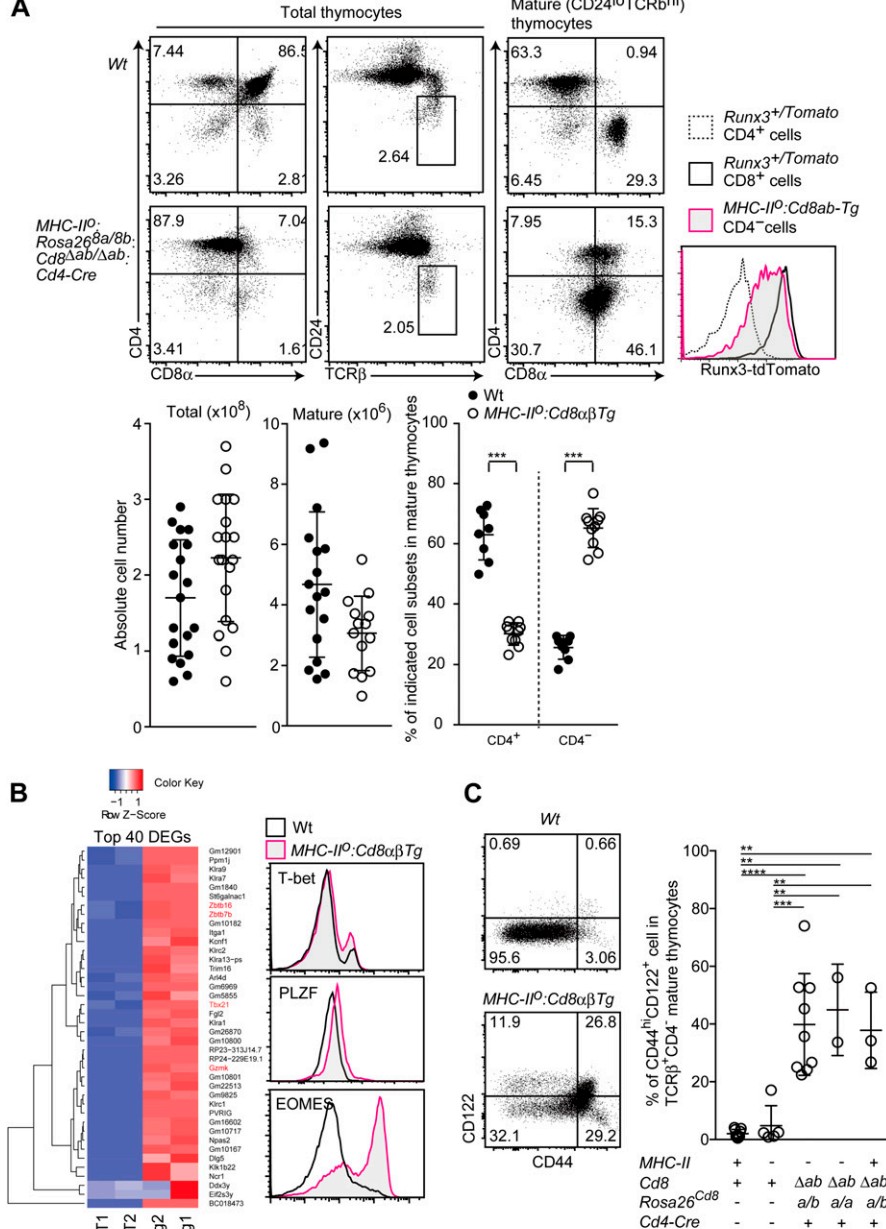

**Figure 3. Differentiation of innate-like CD8⁺ T cell with under constitutive CD8αβ expression.**
**(A)** Flow cytometry analyses for CD4, CD8, TCRβ, CD24, and Runx3-tdTomato expression by various thymocyte subsets of mice with the indicated genotypes. Representative results of more than three independent analyses. Numbers in the plot indicate the percentage of cells in each quadrant. Graphs showing summary of cell numbers of total and mature thymocytes and the percentage of CD4⁺ and CD4⁻ subset in mature thymocytes. Mean ± SD. ***P < 0.001 (unpaired t test, two-sided). **(B)** RNA-seq analyses of CD8 SP cells from indicated genotypes (left). The top 40 differentially expressed genes observed between Wt and Tg are shown. Histograms showing protein expression level of selected genes in indicated genotypes (right). **(C)** Dot plots showing CD44 and CD122 expression in mature CD8 SP thymocyte of the indicated genotypes (left). Representative results of more than three independent analyses. Graph showing summary of percentage of CD44ʰⁱCD122⁺ population in mature thymocytes of indicated genotypes (right). Mean ± SD. **P < 0.01, ***P < 0.001, ****P < 0.0001 (one-way ANOVA with Tukey's multiple comparison).

markers, Plzf and RORγt, that define iNKT2 subset as PlzfʰⁱRORγt⁻ population (Fig S4C). In addition, after ex vivo PMA/ionomycin stimulation, thymic iNKT cells of *MHC-IIᴼ:CD8ab-Tg* mice produced more IL-4 than *wild-type* control mice (Fig 4D), confirming the increase in the iNKT-cell subset that functionally produces IL-4. These observations indicate that constitutive expression of the CD8αβ co-receptor results in a skewed differentiation toward the iNKT2-cell subset.

### Cell-intrinsic and cell-extrinsic mechanisms for innate-like CD8 T-cell development

Our results showed that differentiation of both innate memory-like CD8 thymocytes and the iNKT2-cell subset is enhanced in *MHC-IIᴼ:*

*CD8ab-Tg* mice. Given the known role of IL-4 secreted by iNKT2 cells in supporting innate memory-like CD8 thymocyte differentiation (Verykokakis et al, 2010; Lee et al, 2013), we next tested to what extents the increase in the iNKT2-cell subset has impacts on innate memory-like CD8 T-cell differentiation in *MHC-IIᴼ:CD8ab-Tg* mice. To accomplish this aim, we set up mixed bone marrow chimera experiments in which equal numbers of bone marrow progenitors from CD45.1 *wild-type* mice and CD45.2 *wild-type* or *MHC-IIᴼ: CD8ab-Tg* mice were co-injected together into sublethal irradiated *MHC-IIᴼ* host mice. In the host mice that received CD45.1 wild-type and CD45.2 *wild-type* bone marrow cells, the percentages of CD44⁺CD122⁺ innate memory-like cells in mature CD4⁻CD8⁺ SP thymocytes were similar between CD45.1⁺ and CD45.2⁺ populations

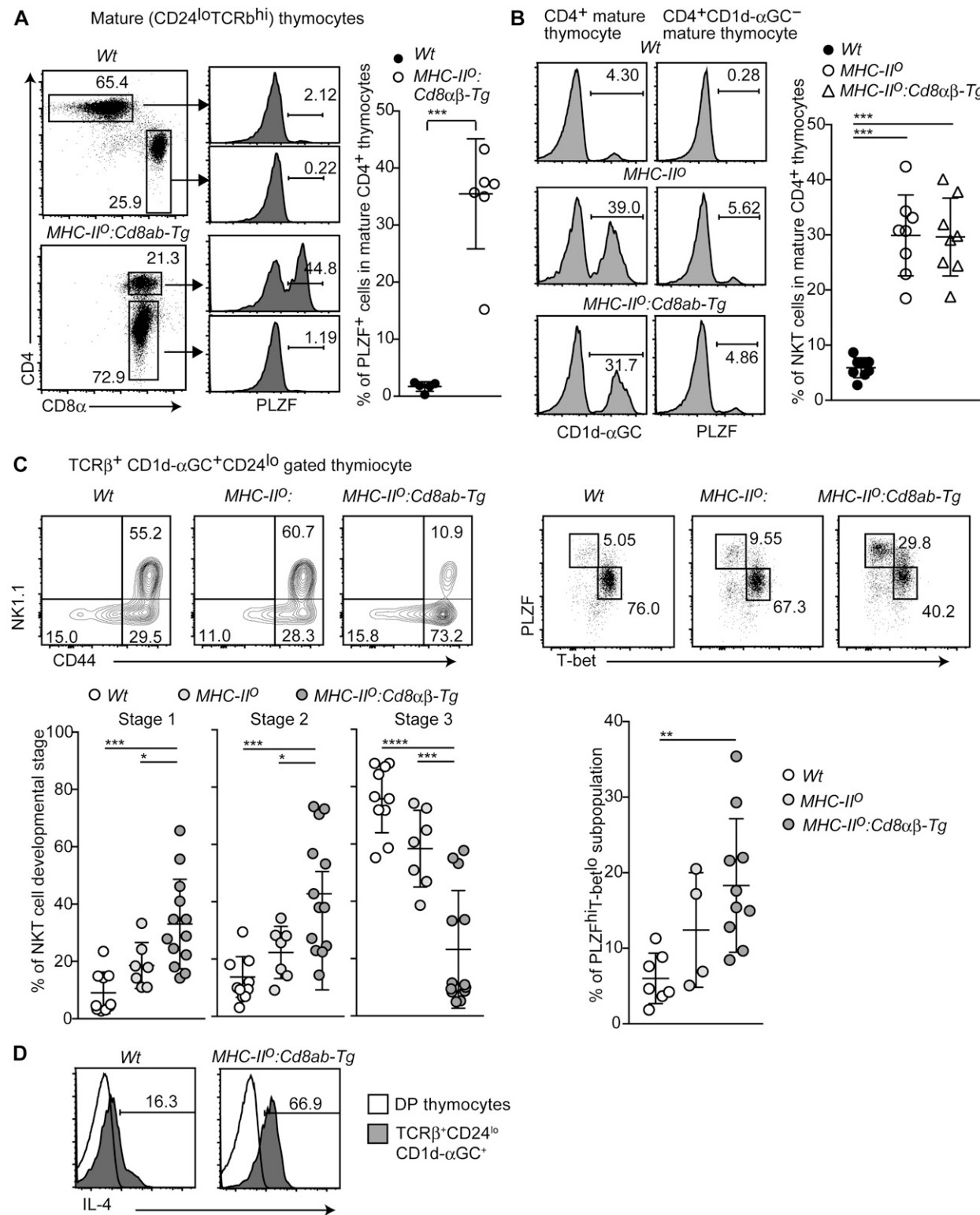

**Figure 4. Skewed differentiation of the iNKT2 subset by constitutive CD8αβ expression.**
**(A)** Dot plots and histograms showing CD4, CD8, and PLZF expression in mature thymocytes of the indicated genotypes (left two panels). Representative results of at least three experiments. Graph showing a summary of five independent experiments (right). Mean ± SD. ***P < 0.001 (unpaired *t* test with Welch's correction, two-sided).
**(B)** Histograms showing CD1d-αGC anti-PLZF staining of the indicated thymocyte subsets and genotypes (left). Graph showing summary of four independent experiments (right). Mean ± SD. ***P < 0.001 (one-way ANOVA with Tukey's multiple comparison). **(C)** Contour and dot plots showing frequencies in various stages and subsets of thymic iNKT cells from the indicated genotypes (upper). CD44⁻NK1.1⁻CD24⁻ stage 1 cells, CD44⁺NK1.1⁻ stage 2 cells, and CD44⁺NK1.1⁺ stage 3 cells. T-bet^lo PLZF^hi is the iNKT2

and were less than 1% in both population (Fig 5A). In striking contrast, the percentage of CD45.1⁺CD44⁺CD122⁺ CD8α⁺ cells were increased to around 5% when progenitors from CD45.2 *MHC-II^O:CD8ab-Tg* mice were included (Fig 5A). The degree of CD45.1⁺CD44⁺CD122⁺ CD8 SP thymocyte differentiation tended to be correlated with the degree of chimerism of CD45.2⁺ cells to CD45.1⁺ cells. These results are consistent with previous studies showing that iNKT2-mediated cell-extrinsic mechanisms, such as an IL-4–enriched thymic environment, support innate memory-like CD8 T-cell differentiation (Verykokakis et al, 2010; Lee et al, 2013). Although actual involvement of IL-4–producing iNKT2-cell subsets in enhanced differentiation of innate memory-like CD8 T cells in *MHC-II^O:CD8ab-Tg* needs to get confirmed by genetics using mice deficient for IL-4 or CD1d, it is likely that skewed differentiation of iNKT2 cells could serve as a cell-extrinsic mechanism driving innate memory-like CD8 T differentiation.

Interestingly, in this setting, the percentage of CD44⁺CD122⁺ cells differentiated from CD45.2 *MHC-II^O:CD8ab-Tg* progenitors was always higher (threefold) than that from CD45.1 *wild-type* cells (Fig 5A). This higher percentage of CD44⁺CD122⁺ cells in the CD45.2⁺ population suggests that besides cell-extrinsic mechanisms such as higher level of IL-4 produced by iNKT2 cells, constitutive CD8αβ expression activates uncharacterized cell-intrinsic mechanisms that makes MHC-I–restricted cells more prone to activate a developmental program toward the innate memory-like CD8 thymocyte.

We next examined development of the iNKT-cell subset in the bone marrow chimera host mice by assessing the ratio of CD45.2⁺ cells to CD45.1⁺ cells at the three stages of iNKT-cell development. This was performed individually for each host mouse because iNKT-cell reconstitution efficiency varied in each host. In the mixed chimera setting of CD45.1 *wild-type* with CD45.2 *wild-type* progenitors, the ratio of CD45.2⁺ cells to CD45.1⁺ cells in each of the three iNKT-cell stages was around one in all host mice (Fig 5B). On the contrary, the proportion of CD45.2⁺ cells relative to CD45.1⁺ cells at the CD44⁻NK1.1⁻ stage 1 and the CD44⁺NK1.1⁻ stage 2 was significantly increased in host mice injected with CD45.1 *wild-type* and CD45.2 *MHC-II^O:CD8ab-Tg* progenitors (Fig 5B). We also confirmed an increased proportion of the T-bet^lo PLZF^hi iNKT2-cell subset in this setting (Fig 5B). These results clearly demonstrate that cell-intrinsic mechanisms are involved in the skewed differentiation toward iNKT1 and iNKT2-cell subsets in *MHC-II^O:CD8ab-Tg* mice, although involvement of cell-extrinsic mechanisms is not formally excluded.

### Intracellular signaling from the CD8α chain is required for iNKT2 skewing

Our results revealed that constitutive CD8αβ expression enhanced iNKT2-cell differentiation in at least a cell-intrinsic manner. Recently, TCR signal strength was shown to be involved in regulating the differentiation of iNKT-cell subsets. Thus, strong TCR signaling promotes iNKT2 and iNKT17 development (Tuttle et al, 2018; Zhao et al, 2018), suggesting that constitutive CD8αβ expression might enhance TCR

signaling during iNKT-cell development, either by aiding in antigen recognition and/or modulating intracellular TCR signals. Consistent with this notion, expression level of CD5, an indicator of TCR signal strength, was higher on thymic iNKT cells that were developed under constitutive CD8 expression (Fig S4D). iNKT cells are selected by CD1d molecules on DP thymocytes (Constantinides & Bendelac, 2013). Contrary to the previous study showing CD1d down-regulation by transgenic CD8 expression (Engel et al, 2010), there was no significant change in CD1d expression levels on DP thymocytes upon transgenic CD8αα or CD8αβ expression (Fig S4E), suggesting that CD1d expression level is unlikely to be involved in enhancement of TCR signaling by transgenic CD8 expression. Two cysteine residues in the CD8α cytoplasmic tail are essential for co-receptor function in TCR stimulation by classical MHC-I via interacting with Lck kinase and recruitment of LAT (Turner et al, 1990; Hoeveler & Malissen, 1993; Bosselut et al, 1999). We, therefore, tested whether these cysteine residues are necessary to enhance iNKT2-cell differentiation by generating a *Rosa26^8aCA* allele that expresses a mutant CD8α chain in which two cysteine residues are replaced with alanine (Fig 6A). Unlike in the *MHC-II^O: CD8ab-Tg* mice, the frequency and numbers of mature CD8 SP thymocytes were significantly decreased in the *MHC-II^O* mice expressing CD8α^CA mutant protein (*MHC-II^O:CD8a^CAb–Tg* mice) (Fig 6B), indicating that co-receptor function supporting conventional MHC-I–restricted cell development was abrogated by these amino acid replacements.

When we examined the iNKT-cell population, defined as CD24^lo TCRβ⁺CD1d-αGC⁺, in *MHC-II^O:CD8a^CAb–Tg* mice, there was no increase in the percentage of the iNKT2-cell subset (Fig 6C). These observations indicate that intracellular signaling through the CD8α chain is necessary to drive enhanced iNKT2-cell development.

It remains unclear whether CD8αβ or CD8αα interacts directly with the non-classical MHC-I molecule CD1d. Nevertheless, our results clearly demonstrated that constitutive CD8α expression alone is sufficient to affect iNKT-cell subset differentiation. We then examined expression kinetics of the endogenous CD8α chain during iNKT-cell development. At the CD24^hi CD44⁻NK1.1⁻ stage 0, most of iNKT-cell precursors express CD8α and CD4 and more than half of these cells down-regulate CD8α expression (Fig S5). During the transition from CD24^hi stage 0 to the CD24^lo CD44⁻NK1.1⁻ stage 1, CD8α expression was efficiently down-regulated. The ThPOK transcription factor is essential to terminate *Cd8* gene expression in conventional MHC-I–restricted cytotoxic T cells (He et al, 2005; Rui et al, 2012) as well as in iNKT cells (Engel et al, 2010). Interestingly, CD8α down-regulation at stage 0 seemed to precede ThPOK induction (Fig S5). Thus, similar to conventional cytotoxic T cells (Muroi et al, 2008), the initial CD8 down-regulation during iNKT-cell development is independent of ThPOK.

## Discussion

Results shown in this study clearly demonstrate that constitutive expression of CD8αβ or CD8αα influences the proportion of iNKT-cell subsets, with skewing to the iNKT2 subset in a cell-intrinsic

subpopulation. One representative result of at least four experiments. Graphs showing summary of at least four experiments (lower). Mean ± SD. *P < 0.05, **P < 0.01, ***P < 0.001 (one-way ANOVA with Tukey's multiple comparison). **(D)** Histogram showing intracellular IL-4 staining in the thymic iNKT population after stimulation with PMA and ionomycin for 4 h. One representative of two experiments.

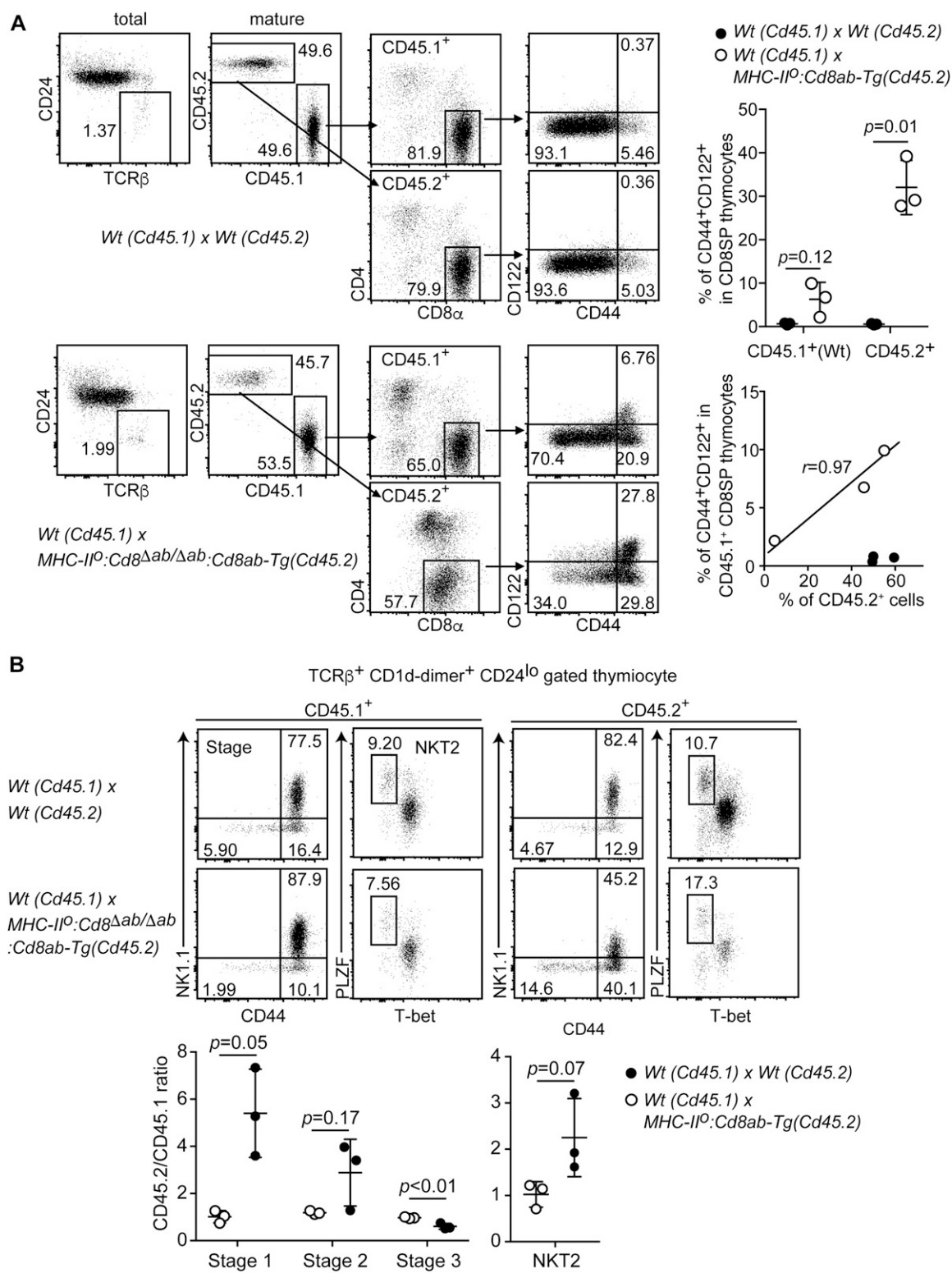

**Figure 5. Cell-intrinsic and cell-extrinsic mechanisms for differentiation of innate-like CD8[+] T cells.**
**(A)** Gating strategy of the flow cytometry analyses for the frequency of the innate memory-like CD8[+] T-cell population in mixed bone marrow chimera experiments. One representative result from three independent mice (left). Graphs showing summary of innate CD8[+] T-cell frequency in the indicated genotypes (right). Mean ± SD. Probability is calculated using two-tailed unpaired *t* test with Welch's correction (right upper panel). The scatter plot indicates the relationships between CD45.2 chimerism and frequency of innate CD8[+] T cells. r indicates Pearson's correlation coefficient. **(B)** Dot plots showing iNKT-cell stages and iNKT2 subpopulations in mixed bone marrow chimera experiments (top). One representative result from three independent mice. Graph showing summary of frequencies of iNKT-cell developmental stages and iNKT2 subpopulation (bottom). Mean ± SD. Indicated probabilities are calculated using two-tailed unpaired *t* test with or without Welch's correction (right upper panel).

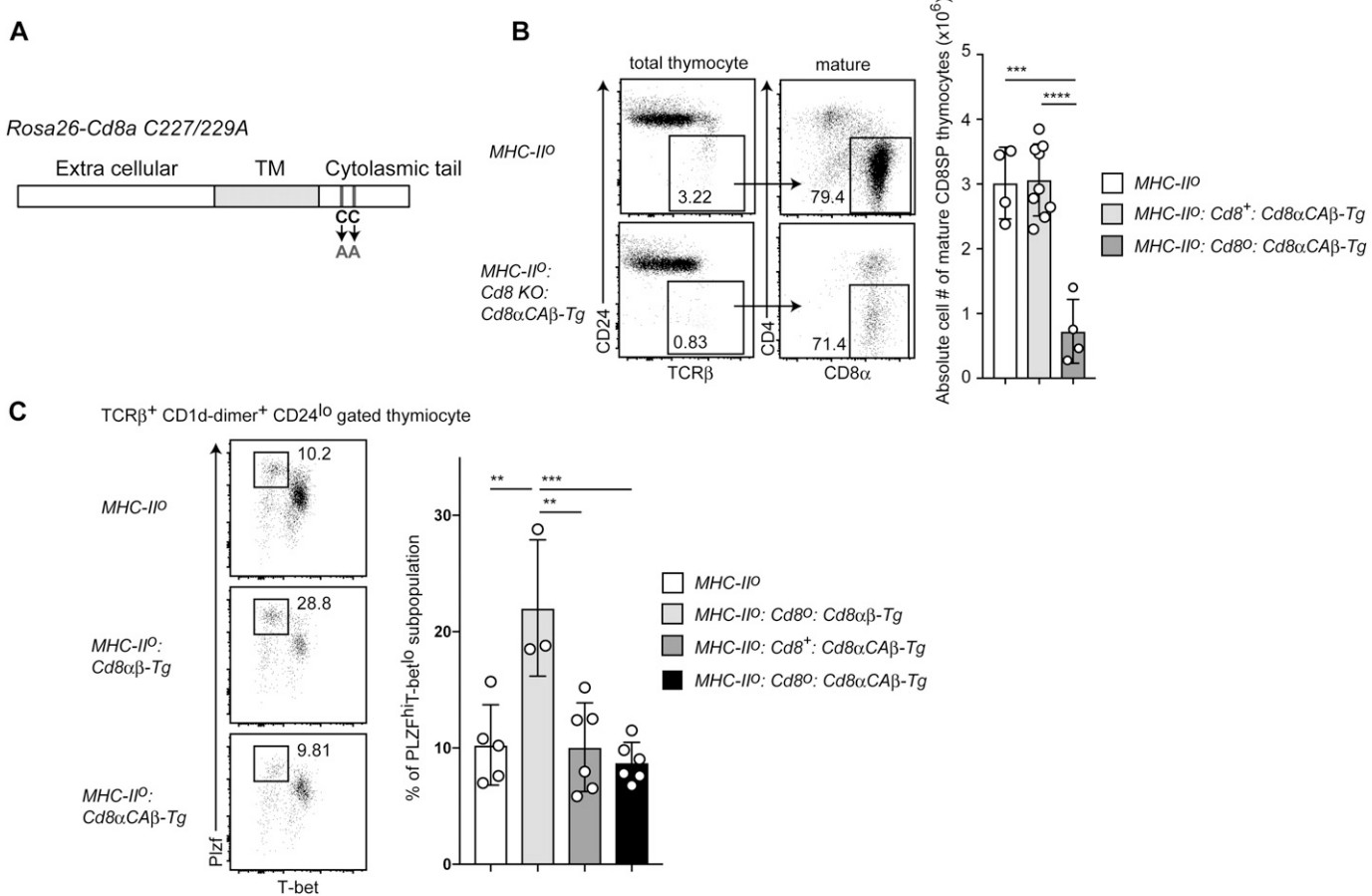

**Figure 6. Signals from CD8α chain are essential for iNKT2 skewing.**
**(A)** Schematic structure of the C227/229A mutant CD8a chain inserted in the *Rosa26* locus. Two cysteine residues at positions 227 and 229 in the CD8α chain cytoplasmic tail were changed to alanine. **(B)** Dot plots showing CD24 and TCRβ expression to define mature thymocytes and CD4 and CD8 expression in mature thymocytes. Numbers in the dot plots indicate percentage of cells in the indicated gates (left). Graph showing summary of absolute number of mature CD8 SP thymocytes of mice with indicated genotype (right). Mean ± SD. ***$P < 0.001$, ****$P < 0.0001$ (one-way ANOVA with Tukey's multiple comparison). **(C)** Dot plot showing T-bet and PLZF expression in thymic iNKT cells of mice of the indicated genotype (top). Numbers indicate percentage of cells in the indicated gate that defines iNKT2 subset. Graph showing a summary of iNKT2 frequency among iNKT cells of mice with the indicated genotype (bottom). Mean ± SD. **$P < 0.01$, ***$P < 0.0001$ (one-way ANOVA with Tukey's multiple comparison). TM, transmembrane domain.

manner, generating an IL-4–rich thymic microenvironment that fosters differentiation of innate memory-like CD8+ T cells. In addition to the cell-extrinsic mechanism, constitutive CD8αβ expression somehow conditions MHC-I–restricted cells to become more prone to acquire innate memory-like signatures in a cell-intrinsic manner.

Recent studies reported that TCR signal strength influences iNKT-cell subset differentiation (Tuttle et al, 2018; Zhao et al, 2018). Attenuated function of ZAP70, which plays an essential role in early TCR signaling events via phosphorylation of LAT, impaired differentiation of iNKT2 and iNKT17, but not iNKT1 cells (Tuttle et al, 2018; Zhao et al, 2018). The expression level of CD5, which parallels TCR signaling intensity, and induction of Nur77, another marker reflecting TCR strength, is higher in iNKT2 cells (Moran et al, 2011; Lee et al, 2013; Tuttle et al, 2018). These findings highlight that strong TCR signaling promotes iNKT2-cell development, at least in part by sustaining Egr2 expression, which results in increases of chromatin accessibility of iNKT2-specific regulatory elements harboring Egr2-

and NFAT-binding motifs (Tuttle et al, 2018). Therefore, it is conceivable that constitutive CD8αβ expression enhances TCR signaling in iNKT precursors. However, a previous study failed to detect direct binding of CD8αβ to the non-classical MHC-I molecule, CD1d (Engel et al, 2010), leaving open the question of how CD8αβ enhances TCR signaling engaged by CD1d. Our results showed that the CD8αα homodimer, which has a distinct affinity for conventional MHC-I and a thymus leukemia antigen (TL), another non-classical MHC-I molecule, from CD8αβ co-receptor (Leishman et al, 2001; Gangadharan & Cheroutre, 2004), also functions to enhance iNKT2 development. This suggests that CD8αβ is unlikely to help iNKT precursors recognize antigen on CD1d as it does for antigens on classical MHC-I as a co-receptor, although the possibility that CD8αβ increases binding affinity of invariant TCR with antigen/CD1d molecules is not formally excluded. Using a retrogenic transgenic mouse system, Cruz Tleugabulova et al showed that TCR half-life is more important to modulate iNKT subset differentiation than avidity (Cruz Tleugabulova et al, 2016). This finding is not only in line

with the idea that CD8$\alpha\beta$ does not increase affinity for antigen/CD1d complexes but also raise the possibility that CD8$\alpha\beta$ stabilizes the interaction between TCR and antigen/CD1d complexes. On the other hands, our results show that two cysteine residues within the intracellular domain of CD8$\alpha$ chain are required for skewed differentiation toward the iNKT2 subset. These cysteine residues function as a docking module for Lck and LAT (Turner et al, 1990; Hoeveler & Malissen, 1993). Such an intracellular association of CD8$\alpha$ chain with TCR/CD3 complexes and signaling molecules in the absence of antigen recognition suggests a possible role of CD8$\alpha$ chain to strengthen TCR signaling independently of extracellular binding of CD8$\alpha$ to CD1d. Given the involvement of ZAP70 in shaping the development of iNKT subsets, the capacity of the CD8$\alpha$ chain to interact with the molecular machinery in early TCR signaling events is likely to be essential to strengthen or prolong TCR signaling. At any rate, given that CD4 is constitutively expressed during iNKT-cell development and interacts with Lck with higher affinity than CD8$\alpha$, it is possible that CD8$\alpha$ modulates intracellular signaling from invariant TCR engaged by antigen/CD1d complexes differently from CD4.

Pioneering studies proposed that CD8$\alpha\beta$ expression during iNKT-cell development inhibits this development through a process of negative selection (Lantz & Bendelac, 1994). However, emergence of CD8[+] iNKT cells in ThPOK-deficient mice indicated that CD8 expression does not provide full inhibitory signals (Engel et al, 2010). Recently, reduction of iNKT-cell numbers in SLAM family receptor–deficient mice was shown to stem from increased apoptosis because of strong TCR signals (Lu et al, 2019). Thus, the SLAM family receptor functions to attenuate TCR signals to foster iNKT-cell development, implying that differentiation of iNKT cells is impaired once TCR signal strength exceeds some thresholds. Therefore, enhanced TCR signaling by constitutive CD8$\alpha\beta$ expression inhibits differentiation of some iNKT precursors that receive excessive TCR signals, whereas other iNKT precursors that can survive under constitutive CD8$\alpha\beta$ expression are directed toward the iNKT2-cell developmental pathway. It is possible that such iNKT2 cells that emerged under constitutive CD8$\alpha\beta$ expression are exposed to prolonged TCR signals and produce an increased amount of IL-4, generating the IL-4–rich thymic microenvironment despite a small increase in their numbers.

A previous publication discussed the possibility that down-regulation of endogenous CD8$\beta$ chain could be what limits the redirection of MHC-I selected cells into CD4[+] helper T cells (Bosselut et al, 2001). However, in the absence of endogenous CD8$\beta$, the proportion of CD4[+] versus CD4[−] cells was about one-third, which is similar to that observed in the previous study (Bosselut et al, 2001), implying that this is not the case. Moreover, one-third of the CD4[+] T cells are CD1d-restricted iNKT cells, and the rest of the CD4[+] T population contains a significant proportion of PLZF[+] innate-like cells with uncharacterized MHC-restriction. Therefore, at least in our experimental setting for transgenic CD8$\alpha\beta$ expression, most MHC-I–selected cells differentiate into CD8[+] cytotoxic-lineage cells. This finding demonstrates that a persistent positive-selection signal alone is not sufficient to instruct the helper-lineage fate to MHC-I restricted cells. Thus, our results challenge the model in which a distinct duration of positive-selection signals between MHC-I–and MHC-II–selected thymocytes generated by distinct

kinetics of expressions of the CD4 versus CD8$\alpha\beta$ co-receptor is central to mechanisms that govern the helper versus cytotoxic lineage choice. Our results also revealed that down-regulation of CD8 plays an essential role in preventing development of innate memory-like CD8 T cells, rather than in inducing conventional cytotoxic fate, at least in part by shaping differentiation of iNKT-cell subsets. By using several experimental systems, disruption of CD4/MHC-II–mediated positive-selection signals had been previously shown to result in a redirected differentiation of MHC-II restricted cells into CD8[+] cytotoxic-lineage cells (Sarafova et al, 2005; Adoro et al, 2012). This genetic evidence supports the concept that persistence of positive-selection signals is necessary to guide MHC-II–selected thymocytes to become the helper-lineage T cells. On the contrary, whether persistence of MHC-I–mediated TCR signaling is sufficient to instruct the helper-lineage fate has not yet been well established. Given the lower expression level of transgenic CD8$\alpha\beta$ in this study, it remains possible that the signal strength was not sufficient to instruct the helper-lineage fate in this setting. However, the ratio of CD4[+] versus CD4[−] mature thymocytes in this study was similar to that observed in mice expressing transgenic CD8$\alpha\beta$ at a level similar to endogenous CD8$\alpha\beta$ (Bosselut et al, 2001). In addition, the lower expression level of transgenic CD8$\alpha\beta$ or CD8$\alpha\alpha$ was sufficient to influence TCR signals from invariant TCR on iNKT precursors. Therefore, it is conceivable that persistence of MHC-I–derived positive-selection signals alone is not sufficient to instruct the helper-lineage fate, although it is still important to test thymocyte fate decision by achieving constitutive CD8$\alpha\beta$ expression at the correct level (Littman, 2016). Our results suggest that there must be uncharacterized differences in positive-selection signals between MHC-I– and MHC-II–restricted thymocytes beyond duration of signal length. Understanding of such differences in positive-selection signals engaged by distinct MHCs is extremely important and awaits experiments in future studies.

The unique kinetics of CD8 expression, known as co-receptor reversal, was discovered 20 yr ago (Brugnera et al, 2000; Cibotti et al, 2000). CD8 co-receptor reversal is controlled at the transcriptional level and is accomplished by temporal termination of *Cd8* gene expression in all post-selection thymocytes and sequential *Cd8* reactivation specifically in the cytotoxic-lineage cells. However, the molecular mechanisms underlying such CD8 expression with its unique kinetics remains uncharacterized. Our results revealed a novel relevance of *Cd8* down-regulation to shaping iNKT-cell subset differentiation, raising the possibility that mechanisms causing *Cd8* down-regulation might be evolved to fine-tune innate T-cell differentiation in species that lack conventional cytotoxic T cells or in the extant positive-selection process of cytotoxic T cells by classical MHC-I molecules. However, expression of the CD8$\alpha\beta$ co-receptor on cytotoxic-lineage cells is beneficial to efficiently mount adaptive immune responses by assisting recognition of peptide antigen presented on MHC-I molecules in the periphery, leading us to speculate that mechanism(s) that reactivate *Cd8* gene might have been acquired later. In mice, the E8I enhancer, one of six enhancers in the *Cd8* locus, shows CD8[+] cytotoxic lineage–specific activity in a reporter transgenic assay (Ellmeier et al, 1997). However, CD8$\alpha\beta$ co-receptor expression by MHC-I–restricted cells is maintained in the absence of the *E8I* enhancer (Ellmeier et al, 1998), indicating that compensatory mechanisms operate to maintain CD8$\alpha\beta$ co-receptor

expression on MHC-I–restricted cytotoxic T cells ([Ellmeier et al, 2002]). Such mechanisms for CD8 expression are likely to have been established under evolutional pressures to secure MHC-I–dependent pathogen clearance.

# Materials and Methods

### Mice

*Thpok^ΔTE* mice ([Muroi et al, 2013]), Rosa26-STOP-YFP mice ([Srinivas et al, 2001]), and Runx3-tdTomato reporter mice ([Kojo et al, 2017]) have been described. I-Aβ–deficient mice were from Taconic. To generate a *Cd8^Δab* allele, we first replaced exon 2 in the *Cd8b1* gene with a neomycin resistant gene (*neo^r*) flanked with FRT sites by homologous recombination with the targeting vector in MI ES cells, establishing ES clone 67. We next removed the *neo^r* gene from the ES clone 67 by transient transfection of an FLP recombinase expression vector and isolated an ES clone harboring the *Cd8^Δb* allele, into which the targeting vector used to generate the *Cd8a^h2PA* allele ([Wada et al, 2018]) was transfected to replace exon 1 of the *Cd8a* gene with an hCD2 cDNA and the *neo^r* gene. After isolation of ES clones that had undergone homologous recombination with the second targeting vector at the *Cd8a* gene, we screened them to determine which allele, *Cd8* or *Cd8^Δb*, was targeted by analyzing the type of FLP recombinase-mediated recombination occurring after transduction of a retrovirus vector encoding FLP, and then isolated ES clones harboring the *Cd8^+/ΔbaCD2N* genotype. After removal of the *neo^r* gene from the *Cd8^+/ΔbaCD2N* ES clone by transient transfection of the FLP recombinase expression vector, ES clones were used to generate chimera mice by aggregation. To generate the *Rosa26^8a* or *Rosa26^8b* allele, a cDNA fragment encoding CD8α or CD8β was PCR-amplified by RT-PCR using thymus mRNA, sequenced, and cloned into the *AscI* site of the CTV vector (#159212; Addgene). A cDNA encoding the mutant CD8α^CA chain harboring alanine substitutions at two cysteine residues in the CD8α cytoplasmic tail was amplified by overlap PCR. All mice were maintained in the specific pathogen-free animal facility at the RIKEN IMS, and all animal procedures were in accordance with institutional guidelines for animal care and with the protocol (28-017) approved by the Institutional Animal Care and Use Committee of RIKEN Yokohama Branch.

### Flow cytometry and cell sorting

Single-cell suspensions from the thymus, spleen, and lymph nodes were prepared by mashing tissues through a 70-μm cell strainer (BD Bioscience). Single-cell suspensions were stained with the following antibodies purchased from BD Bioscience, eBiosciences, or BioLegend: CD4 (RM4-5), CD8a (53-6.7), CD24 (M1/69), CD44 (IM7), CD122 (TM-β1), TCRβ (H57-597), NK1.1 (PK136), CD69 (H1.2F3), and CCR7 (4B12). Murine CD1d-dimers X I (557599; BD Biosciences) obtained from BD Biosciences were loaded with αGalCer (KRN7000; Funakoshi) and were labelled with antimouse IgG1 proximal enhancer (550083; BD Biosciences) or antimouse IgG1 BV510 (740421; BD Biosciences). For intracellular staining of transcription factors, the

cells were stained with cell surface molecules, and then fixed and permeabilized using the Transcription Factor Buffer Set (562574; BD Biosciences). Permeabilized cells were stained with antibodies from BD Biosciences, eBiosciences, or BioLegend: T-bet (4B10), PLZF (9E12), Eomes (Dan11mag), and ThPOK (T43-94). To stain intracellular IL-4, total thymocytes were stimulated with 50 ng/ml in ml of phorbol 12-myristate 13-acetate (PMA) (Sigma-Aldrich) and 1.5 μM of ionomycin for 4 h in the presence of BD Golgi Stop (554724; BD Biosciences). Then, the cells were stained with cell surface molecules and were fixed and permeabilized using BD Cytofix/Cytoperm Kit (554714; BD Biosciences). Permeabilized cells were stained with anti-IL-4 antibody (11B11; BD Biosciences). Multi-color flow cytometry analysis was performed using a BD FACSCanto II (BD Bioscience), and data were analyzed using FlowJo (BD Bioscience) software. Cell subsets were sorted using a BD FACSAria II or III (BD Biosciences).

### Mixed bone marrow chimera

Recipient MHC-II-deficient mice were sublethally irradiated at 950 rad and were reconstituted with $5 \times 10^6$ bone marrow cells from CD45.2^+ wild-type mice or CD45.2^+ *MHC-II^O*:CD8αβ Tg mice with $5 \times 10^6$ CD45.1^+ bone marrow cells. At 8–12 wk after transfer, the spleen and thymus were analyzed by flow cytometry.

### Quantitative RT-PCR

Total cellular RNA was extracted from purified cell subsets using Trizol (Thermo Fisher Scientific) and treated with RNase-free DNase I (Thermo Fisher Scientific). cDNA was synthesized from total RNA using the SuperScriptII First Strand Synthesis System (Invitrogen). Quantitative RT-PCR was performed using the ABI/PRISM 7000 sequence detection system with an internal fluorescent TaqMan probe. Primers for *Zbtb7b* ([He et al, 2005]) and *Eomes* ([Intlekofer et al, 2005]) were previously described.

### RNA-seq

RNA was extracted from sorted CD8^+ and CD4^− mature thymocytes of *Wt* and *MHC-II^O*:CD8αβ Tg mice by TRIzol (Thermo Fisher Scientific), followed by RNeasy Micro Kit (QIAGEN) according to the manufacturer's protocols. Sequencing libraries were prepared using a NEBNext RNA Library Prep Kit for Illumina (E7530; NEB) with Poly(A) mRNA Magnetic Isolation Module (E7490; NEB) according to the manufacturer's protocol. Single-end 50-bp reads were obtained by Illumina HiSeq 1500. The reads were mapped by Tophat v2.1.1 onto the mouse genome mm10 and counts were generated using HTSeq v.0.6.1. The DESeq2 R package was used to perform principle component analysis and to generate the heat map of differentially expressed gene with normalized counts of *wild-type* ([GSE48138]) and *Tg* ([GSE132059]) in duplicates in this study.

### Accession codes

GEO: raw sequencing data and processed files, [GSE48138] and [GSE132059].

# Supplementary Information

# Acknowledgements

We are grateful to T Ishikura for aggregation of ES cells, Y Taniguchi for mouse genotyping, N Yoza for cell sorting, and Dr Wilfried Ellmeier for critical reading of the manuscript. This work was supported by the Grants-in-Aid for Scientific Research (B) (19390118) from JSPS and the Grants-in-Aid for Scientific Research on Innovative Areas (17H05805 and 19H04820) from the MEXT in Japan (I Taniuchi).

## Author Contributions

S Kojo: formal analysis and investigation.
M Ohno-Oishi: formal analysis and investigation.
H Wada: formal analysis and investigation.
S Nieke: methodology.
W Seo: formal analysis.
S Muroi: methodology.
I Taniuchi: formal analysis and writing—original draft.

## Conflict of Interest Statement

The authors declare that they have no conflict of interest.

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
