## [Reviewer comments · Life Science Alliance]

Life Science Alliance

Constitutive CD8 expression drives innate CD8+ T cell differentiation via induction of iNKT2 cells.

Satoshi Kojo, Michiko Ohno-Oishi, Hisashi Wada, Sebastian Nieke, Wooseok Seo, Sawako Muroi and Ichiro Taniuchi

DOI: <https://doi.org/10.26508/lsa.202000642>

Corresponding author(s): Ichiro Taniuchi

Review Timeline:	Submission Date:	2020-01-08
	Editorial Decision:	2020-01-14
	Revision Received:	2020-01-16
	Accepted:	2020-01-17

Scientific Editor: Andrea Leibfried

Transaction Report:

Please note that the manuscript was previously reviewed at another journal and the reports were taken into account in the decision-making process at Life Science Alliance. Since the original reviews are not subject to Life Science Alliance's transparent review process policy, the reports and author response cannot be published.

January 14, 2020

RE: Life Science Alliance Manuscript #LSA-2020-00642-T

Dr. Ichiro Taniuchi
RIKEN, Center for Integrative Medical Sciences (IMS)
1-7-22, Suehiro-cho, Turumi-ku
Yokohama, Kanagawa 230-0045
Japan

Dear Dr. Taniuchi,

Thank you for transferring your revised manuscript entitled "Constitutive CD8 expression drives innate CD8+ T cell differentiation via induction of iNKT2 cells" to Life Science Alliance.

Your manuscript was reviewed and re-reviewed at another journal before, and the editors transferred those reports to us with your permission.

Upon re-review, reviewer #1 appreciated the revision but pointed out that the mechanistic understanding of your findings remains low. Reviewer #2 also thought that the mechanistic insight is limited and that her/his comment regarding iNKT cells in wild-type mice was not sufficiently addressed. Reviewer #3 was happy with the revisions performed. A new reviewer #4, involved only upon re-review, thought that insight into effects on thymus size in the transgenic mice is needed. This reviewer found the work preliminary, asked how cell numbers as compared to the reported cell % are affected and raised some other concerns.

We have assessed your work in light of all remaining comments for consideration at Life Science Alliance. Absence of further mechanistic insight does not preclude publication here and we also think that more insight into the iNKT cells in wild-type mice is not needed at this stage. Furthermore, the new points raised by reviewer #4 do not need addressing for the following reasons: First, these concerns were not raised in the initial round of review of your work and the original reviews seemed thorough. Second, we see value in your model, so the concern regarding artificiality and complexity of the system can get overruled. Third, the valid question regarding cell numbers versus frequencies (%) can get discussed in our view within the manuscript text as both seem equally important in a developmental context and we appreciate the high variability in cell numbers that can be found in the thymus. Fourth, the dependency of the effect on IL-4 can in our view get reasonably assumed, so this reviewer concern can get addressed by further discussion/softening the language, too.

We would thus like to invite you to submit a final version of your manuscript for publication in Life Science Alliance, a version that includes the following revisions:

- Expand the discussion on cell number vs frequencies as mentioned above
- Expand the discussion on IL-4 dependency as mentioned above;
- obviously, you can also still include any other revisions or text changes in response to the remaining reviewer concerns at this stage
- Please confirm that all mouse experiments were performed in accordance with relevant guidelines and regulations. The manuscript must include a statement in the Materials and Methods section

identifying the institutional and/or licensing committee approving the experiments.

- Please upload the supplementary figures as individual files and without legends; the legends should get moved into the main manuscript docx file
- Some of your figures are in landscape format, please consider re-formatting Fig 3 and 4 into portrait format to allow for a larger display of the data within the pdf and HTML version of the paper
- Please provide a 'summary blurb' within our submission system and fill-in the electronic license to publish form

A. FINAL FILES:

B. MANUSCRIPT ORGANIZATION AND FORMATTING:

****It is Life Science Alliance policy that if requested, original data images must be made available to**

the editors. Failure to provide original images upon request will result in unavoidable delays in publication. Please ensure that you have access to all original data images prior to final submission.**

The license to publish form must be signed before your manuscript can be sent to production. A link to the electronic license to publish form will be sent to the corresponding author only. Please take a moment to check your funder requirements.

Thank you for your attention to these final processing requirements.

Sincerely,

January 17, 2020

RE: Life Science Alliance Manuscript #LSA-2020-00642-TR

Dear Dr. Taniuchi,

Thank you for submitting your Research Article entitled "Constitutive CD8 expression drives innate CD8+ T cell differentiation via induction of iNKT2 cells.". It is a pleasure to let you know that your manuscript is now accepted for publication in Life Science Alliance. Congratulations on this interesting work.

DISTRIBUTION OF MATERIALS:

Again, congratulations on a very nice paper. I hope you found the review process to be constructive and are pleased with how the manuscript was handled editorially. We look forward to future exciting submissions from your lab.

Sincerely,

Andrea Leibfried, PhD
Executive Editor
Life Science Alliance
Meyerhofstr. 1

69117 Heidelberg, Germany
t +49 6221 8891 502
e a.leibfried@life-science-alliance.org
www.life-science-alliance.org